# Oculometric Assessment of Sensorimotor Impairment Associated with Liver Disease Is as Sensitive as Standard of Care Cognitive Tests

**DOI:** 10.3390/geriatrics10040112

**Published:** 2025-08-19

**Authors:** Dorion Liston, Katherine Wong, Aaron Yeoh, Shalonda Haywood, Aparna Goel, Paul Kwo, Quinn Kennedy, Philip N. Okafor

**Affiliations:** 1neuroFit, Inc., Mountain View, CA 94040, USA; qkennedy@neurofit.tech; 2Division of Gastroenterology and Hepatology, Department of Medicine, Stanford University School of Medicine, Stanford, CA 94305, USAgoela21@stanford.edu (A.G.); pkwo@stanford.edu (P.K.); 3Department of Medicine, Stanford University School of Medicine, Stanford, CA 94305, USA; ayeoh@stanford.edu; 4Stanford Healthcare, Stanford, CA 94305, USA; shaywood@stanfordhealthcare.org; 5Department of Gastroenterology and Hepatology, Mayo Clinic, Jacksonville, FL 32224, USA; philokafor@post.harvard.edu

**Keywords:** sensorimotor, liver disease, diagnostic tool, pursuit eye movement, neurocognitive impairment

## Abstract

Significance: Hepatic encephalopathy (HE) occurs in 20–80% of patients with liver cirrhosis, impacting attention, memory, processing speed, and visuospatial skills. HE standard-of-care psychometric assessments are time-consuming and require staff training. Oculometrics may provide a fast, non-invasive brain health assessment that can be self-administered in a medical environment. Purpose: We investigated whether an oculometric assessment could measure the severity of HE as accurately as standard-of-care psychometric methods. Methods: Forty-eight participants (19 with decompensated cirrhosis, 10 with compensated cirrhosis, 19 controls) completed a previously validated five-minute oculometric test and the standard-of-care psychometric hepatic encephalopathy (PHE) battery. The oculometric test consists of following a dot as it moves across a computer screen and generates 10 metrics including a summary score called nFit. The PHE battery entails five standard cognitive tests, generating seven metrics including a PHE composite score (PHES). Results: The oculometric summary score, nFit, correlated with the current diagnostic standard, the PHES (*r* = 0.51, *p* < 0.001), the presence or absence of HE as determined by PHES composite (*r* = −0.44, *p* < 0.001), as well as the severity of cirrhosis (*r* = −0.59, *p* < 0.001). Additionally, performance on both nFit and PHES distinguished compensated (ROC: nFit: 0.71, PHES: 0.68) and decompensated (ROC: nFit: 0.88, PHES: 0.85) patient groups from control participants comparably. Finally, compared to participants with decompensated cirrhosis, control participants had better scores for almost all oculometrics: acceleration, catch-up saccade amplitude, proportion smooth, direction noise, and speed noise. Conclusions: Patients with liver disease showed impairment on multiple aspects of visual processing compared to a control group. These functional visual processing impairments correlate with the presence or absence of HE, showing significant sensitivity in distinguishing people with HE from controls. Oculometric tests provide a quick, non-invasive functional assessment of brain health in patients with liver disease, with sensitivity indistinguishable from standard-of-case psychometric tests.

## 1. Introduction

Hepatic encephalopathy (HE) is a complication of liver disease defined by a spectrum of neurocognitive signs and symptoms, leading to reduced quality of life [1] and elevated risk of adverse outcomes [2,3]. HE occurs when the progression of liver disease goes from compensated (when the liver can still function adequately) to decompensated (live failure begins). Decompensated liver disease is marked by multiple complications, including HE. Liver disease is defined by cirrhosis, scarring of the liver that impedes its ability to metabolize and filter out toxins, leading to a build-up of ammonia in the blood which interacts with inflammatory processes [4] to degrade brain function. Due to these interactions, laboratory blood tests do not predict the presence or absence of neurocognitive impairment and standard-of-care psychometric tests [5] are infrequently adopted in clinical practice [1,6]. The liver disease literature cites the need for a readily available, noninvasive clinical tool to quickly and quantitatively screen for neurocognitive impairment associated with HE [1,7,8,9,10].

We replicated the methods of a previous study of traumatic brain injury [11] to assess the feasibility of using an oculometric assessment to screen for signs of HE in patients with cirrhosis. The term “oculometric” [12] refers to eye-movement-based methods to quantify aspects of vision and brain health, from relatively low-level physiological metrics like latency [13], acceleration, and saccadic peak velocity, up to higher-level metrics that quantify the subject’s percept of the stimulus, including direction [14], speed [15], contrast [16], and brightness [17], using techniques adapted from psychophysics [18]. Current oculometric methods involve standardized tasks [19], standardized hardware and software [20], and normative populations [21,22] to relate any pattern of sensorimotor performance to an extensive catalog of normative and pathological oculomotor signs [23], including Mild Cognitive Impairment, early-stage dementia, and traumatic brain injury [11,24,25]. While the few eye-movement studies in patients with cirrhosis report prolonged saccade latencies [11] and smooth pursuit disruptions [12], the characterization of sensorimotor deficits remains incomplete and has not been developed into a screening or diagnostic tool.

In this study, we measured oculometric performance in three patient groups: those with compensated cirrhosis, those with decompensated cirrhosis, and healthy controls. We compared those results to clinically validated neurocognitive assessments [7,26,27,28]. We hypothesized that oculometrics would be as good as the current diagnostic standard of HE related neurocognitive impairment in distinguishing neurocognitive function between the three groups. We also hypothesized that oculometric performance would correlate with performance on the standard neurocognitive test battery.

## 2. Methods

### 2.1. Participants

Twenty-nine adults with cirrhosis (19 decompensated, 10 compensated) and 19 healthy controls were recruited for this feasibility study; further demographic details are given in Table 1. Potential participants were identified by weekly review of the Division of Gastroenterology and Hepatology’s pretransplant liver and gastroenterology clinic schedules. Liver disease is defined by cirrhosis, scarring of the liver, diagnosed using imaging and/or liver biopsy along with clinical and laboratory correlation. Patients with compensated and decompensated cirrhosis and noncirrhotic controls were considered for enrollment after obtaining Stanford University IRB approval. Exclusion criteria included: (i) individuals under 18 years of age, (ii) individuals with uncontrolled neuropsychiatric illnesses or overt hepatic encephalopathy, (iii) ongoing alcohol or illicit substance use, (iv) non-English speakers because the PHES has not been validated among non-English speaking patients in the United States, and (v) individuals who have undergone liver transplant. Some patients with decompensated cirrhosis had a past diagnosis of overt HE (*n* = 9 of 19 patients), which is a diagnosis of exclusion applied after factors other than liver disease have been ruled out [1].

### 2.2. Measures

Model for End-stage Liver Disease (MELD) [29]. This calculated formula measures mortality risk in patients with end-stage liver disease. Scores range from 6 to 40; higher scores indicate greater risk of mortality. Child-Turcotte-Pugh (CTP) [30]. This scoring system measures mortality risk in cirrhosis patients. Scores range from 5 to 15, with 5–6 points indicating good hepatic function, 7–9 points indicating moderate impairment, and 10–15 indicating advanced hepatic dysfunction. Psychometric hepatic encephalopathy (PHE) battery [10]. The paper-based PHE battery entails five standard cognitive tests: Number Connection Test A (NCT-A), Number Connection Test B (NCT-B), Line Tracing Test (LTT), Digital Symbol test (DST), and the Serial Dotting Test (SDT). Performance on the NCT-A, NCT-B, LTT, and SDT is measured in seconds needed to complete the task. DST performance is measured by the number of digit symbols successfully completed in 120 s. LLT is measured by the number of errors made and a composite score (LTT seconds plus number of errors). Multiple studies have also used a PHE composite score (PHES) [31,32]. Permission to use the PHES [33] was obtained from Dr. Karin Weissenborn, University of Hannover, Germany. nFit oculometric assessment [22]. We used a standardized, automated five-minute assessment of dynamic vision adapted from the Rashbass (1961) step-ramp tracking task [34] described in detail previously [22]. The task is performed on a desktop device (described below). On each of 45 trials, a dot is presented in the middle of the screen, and at a random time, it moves in a random direction at a random speed. Subjects are instructed to follow the dot for as long as it is visible with their eyes. Specifically, the subject fixates on a central (0.5 deg diameter) black dot and initiated the trial with a button press. Following a randomized interval (truncated exponential distribution, mean: 700 ms, range: 200–5000 ms), the dot made a small position step (displacement of 200 ms of target motion) in a random direction around the unit circle and began moving back toward the initial fixation location at a random speed (16–24 deg/s). The task generates 10 z-scored metrics, including a summary score called nFit: acceleration, gain, catch-up saccade amplitude, proportion smooth tracking, direction noise, speed tuning responsiveness, and speed noise. Detailed information about the computation of these metrics can be found in Liston & Stone, 2014 [22].

### 2.3. Equipment

We used neuroFit eye-tracking hardware (neuroFit, Mountain View, CA, USA), an oculometric assessment device designed for use in the clinic to display the stimulus sequence for any standardized oculomotor protocol [19,21,22,35], collect high-quality monocular or binocular eye-position data, and compute summary metrics. The non-invasive eye-tracking system [20] collects high-precision 2D eye-position signals (noise level of 0.2 deg in human control subjects, <0.01 deg for a glass eye) using two 850 nm IR illuminators and a central image sensor. Using a custom-designed ergonomic chinrest (neuroFit, Mountain View, CA, USA), the subject’s head is stabilized at a viewing distance of 57 cm from the LED-backlit LCD display (24″ display, 1920 horizontal, 1080 vertical, 120 Hz). Using an automated module for detection and registration of facial features, the system can track one or both eyes for any subject within the 5.35–7.25 cm anthropometric range of inter-pupillary distances [36]. In the present study, we used 250 Hz monocular tracking of the left eye.

### 2.4. Procedures

Consenting participants were surveyed regarding their level of education, history of falls and motor vehicle accidents (MVA) while driving in the past year, and history of hospitalization for HE in the past year. They then performed the paper and pencil PHE battery followed by the 5 min oculometric assessment of dynamic visual processing [22].

### 2.5. Statistical Analyses

We extracted data from patient charts including demographics, medications, medical comorbidities, and laboratory data to calculate the MELD and CTP scores. These scores were used to classify each participant into one of the three groups (compensated cirrhosis, decompensated cirrhosis, and controls). An a priori power calculation using G*Power 3.1.9.7 [37] for a one-way ANOVA with an alpha level of 0.05, power of 0.80 indicated that a total sample size of 42 is required. Thus, our sample size of 48 is adequate to test the study hypothesis.

Demographic and clinical comparisons between the three groups were made using descriptive statistics (Table 1). Table 2 provides the descriptive statistics of PHE battery and oculometric performance across the three groups. We performed Kruskal–Wallis tests to determine if PHE and oculometric performance significantly differed across groups (see Table 2). Using the categorical identification of each participant (control, compensated cirrhosis, decompensated cirrhosis) provided by the Division of Gastroenterology and Hepatology, we performed Receiver Operating Characteristic (ROC) analysis to quantify the sensitivity of the psychometric and oculometric assessments to detect the presence or absence of neurocognitive signs of liver disease. Finally, we computed correlations between the oculometric and PHE measures. A minimal dataset of the de-identified raw data is provided as Appendix A.

## 3. Results

### 3.1. Demographic Data

Twenty-nine adults with cirrhosis (19 decompensated, 10 compensated) and 19 healthy controls were recruited for this feasibility study; further demographic details are given in Table 1. Within our cirrhosis cohort, median MELDNa was 12 (SD 4.35, range 6–24) and CTP score was 6 (SD 1.34, range 5–10). There was a male predominance (66%), with higher BMI, fewer years of education, and the most common etiologies were: hepatitis C (38%), nonalcoholic steatohepatitis (24%), and alcohol (14%). Of the participants with decompensated cirrhosis, 15 of 19 (79%) had been diagnosed with overt HE in the past, and 7 of 19 (37%) had been hospitalized for overt HE in the past year. Of the patients with an overt HE diagnosis, 79% were managed with lactulose alone and 13 of 19 (68%) were managed with both rifaximin and lactulose. Compared to participants with cirrhosis, the control group had more years of education (16.8 vs. 14.6 years, *p* < 0.01), were younger (46.5 vs. 59.7 years, *p* < 0.01), and had lower BMI (23.4 vs. 29.2, *p* < 0.05).

To assess the relative contribution of the three demographic variables in our dataset on neurocognitive function (age, BMI, and years of education), we ran independent correlations (Pearson’s R) against nFit and the PHES composite. For nFit, we observed significant correlation with age (*r* = −30, *p* < 0.05), BMI (*r* = −0.37, *p* < 0.01), years of education (*r* = 0.45, *p* < 0.01). For the PHES composite, we observed a significant correlation with years of education (*r* = 0.48, *p* < 0.001), a weak correlation with age (*r* = −0.24, *p* = 0.09) but not with BMI (*r* = −11, *p* = 0.21). Within the control group, nFit retained the effect of BMI (*r* = −0.59, *p* < 0.01), and years of education (*r* = 0.50, *p* < 0.05) but not age (*r* = −0.35, *p* = 0.15), whereas PHES showed no effect of age (*r* = −0.28, *p* > 0.10), BMI (*r* = 0.05, *p* > 0.10) or years of education (*r* = −0.07, *p* > 0.10).

### 3.2. Oculometric and PHES Between Groups Comparisons

Performance on the PHE and oculometric tests showed the same pattern of group differences, in which, on average, patients with decompensated cirrhosis had the worst performance and controls had the best performance. As shown in Table 2, group differences were significant for almost all variables. Figure 1 illustrates the oculometric measurements of a representative control and a patient with HE, showing the relatively sluggish responses and disorderly tracking of the patient with HE. Across the three groups, oculometric impairment was most pronounced in proportion smooth tracking and direction noise, both showing individual sensitivity to distinguish participants with decompensated cirrhosis from control participants (ROC > 0.85, Bonferroni-corrected Bootstrap test, *p* < 0.01) [Figure 2].

Both psychometric and oculometric scores showed degradation as cirrhosis advanced (Figure 3), sharing significant underlying variance (Figure 4A, *r* = 0.50, *p* < 0.001). To compare the sensitivity of these two approaches to distinguish between control patients, patients with compensated cirrhosis, and patients with decompensated cirrhosis, we plotted the histograms of test scores for the three groups (Figure 4B–E). For both the PHES composite score and nFit, we observed overlap between the categorical assignments of our subjects. The ability of the PHES composite score and nFit to distinguish control patients from those with cirrhosis was statistically indistinguishable (Bootstrap test, *p* > 0.05) for both our compensated (PHES 5th, 50th, 95th percentile ROC areas: 0.49, 0.69, 0.85; nFit 5th, 50th, 95th percentile ROC areas: 0.53, 0.71, 0.86) and decompensated (PHES 5th, 50th, 95th percentile ROC areas: 0.74, 0.86, 0.94; nFit 5th, 50th, 95th percentile ROC areas: 0.77, 0.88, 0.97) groups.

Because several demographic variables were correlated with nFit and PHES composite score, we ran regressions analyses in which age, BMI, and years of education were included as covariates to the model. Results indicated that only cirrhosis diagnosis was a significant predictor of nFit score (*b*_1_ = −0.80 *SE* (*b*_1_) = 0.31), *t* = −2.61, *p* = 0.01). For the PHES composite score, the intercept (*b*_0_ = −10.99, *SE* (*b*_0_) = 4.13, *t* = −2.66, *p* = 0.01), cirrhosis diagnosis (*b*_1_ = −2.5, *SE* (*b*_1_) = 0.69, *t* = −3,64, *p* < 0.001) and years of education (*b*_4_ = 0.44 *SE* (*b*_4_) = 0.17, *t* = 2.65, *p* = 0.01) were significant.

### 3.3. Oculometric Correlations with PHES

nFit correlated with the PHES composite score (Spearman’s *r* = 0.51, *p* = 0.0002) (see Figure 4A), as well as the NCT-A (Spearman’s *r* = −0.48, *p* = 0.0005), NCT-B (Spearman’s *r* = −0.48, *p* = 0.0006), DST (Spearman’s *r* = 0.68, *p* < 0.0001), LTT errors (Spearman’s *r* = −0.48, *p* = 0.0006), and the SDT (Spearman’s *r* = −0.38, *p* = 0.007). In sum, longer completion times for the NCT-A, NCT-B, and SDT and more errors on the LTT were associated with worse nFit scores, whereas stronger performance on the DST was associated with better nFit scores.

## 4. Discussion

The consequences of impaired cognition and sensorimotor processing in patients with cirrhosis have raised the need for better and more accessible screening tests to improve early detection and intervention, leading to better quality of life. In this cross-sectional feasibility study, we compared psychometric and oculometric data from 19 control participants, 10 patients with compensated cirrhosis, and 19 patients with decompensated cirrhosis. The oculometric assessment was easy to administer and simple for patients to follow. Our results revealed that patients with decompensated cirrhosis score lower for almost all components of PHES and oculometric testing compared to control patients. Impairments in dynamic visual processing measured with oculometric methods correlated with functional impairments measured with paper-and-pencil psychometric tests, showing significant sensitivity in distinguishing patients with different stages of liver disease from control patients.

We observed some demographic differences between our patients with cirrhosis and control patients. The patients with cirrhosis had a higher BMI (29.2 vs. 23.4) but we attributed this to the fact that 52% of these patients in the study had ascites. The control patients were younger (46.5 vs. 59.7 years, *p* < 0.01) and had more years of education (16.8 vs. 14.6 years, *p* < 0.01), which correlated with both psychometric and nFit scores (*p* < 0.01). In addition to demonstrating the presence of strong cirrhosis-related oculometric signals, this early feasibility study has allowed exploration of subtle physiological factors within our groups that may contribute to degradation in visual tracking. For the two least impaired categories, control and compensated cirrhosis, the relationship between nFit and factors related to BMI (e.g., fatty liver, body fat) also shows a measurable signal that we are currently further exploring. Although the present study was not designed to assess the contribution of BMI-related factors to nFit independent of the patient’s categorical liver-disease diagnosis, the presence of these trends in the two least-severe groups suggests that the early mechanisms of functional neurological impairment may involve factors linked to body weight or systemic inflammation, consistent with previous psychometric [38] and immunological [4] results.

Based on ROC analysis of our oculometric data, we found a statistically significant difference between control patients and patients with decompensated cirrhosis for several metrics, especially the proportion of smooth pursuit versus saccades and direction noise. Compared to the PHES, nFit had a slightly higher sensitivity in distinguishing patients with decompensated cirrhosis from control. These findings indicate that patients with cirrhosis at any stage have significant sensorimotor/psychomotor impairments (e.g., prolonged latency, sluggish acceleration), impaired perception of object motion (e.g., low gain, elevated direction and speed noise, muted responsiveness to changes in speed), and a relative inability to visually track a moving object smoothly (e.g., relatively low proportion of smooth pursuit tracking, larger catch-up saccade amplitude). Bajaj et al. also demonstrated that patients with advanced cirrhosis with a prior episode of HE also had worse cognitive performance in regard to both speed and accuracy on the Stroop test [39]. While our current sample size is too small to test for links between deficits in visuomotor processing and adverse outcomes in the HE group (e.g., falls and motor vehicle accidents), a future study with a larger sample and a longitudinal follow-up will allow these hypotheses to be tested.

Our feasibility study did have some limitations that temper generalizability of the results. First, our samples of decompensated cirrhosis, compensated cirrhosis, and control patients were not balanced for demographic variables. The patients with cirrhosis were older, with higher BMI, and fewer years of education. While these demographic variables were individually correlated with nFit and PHES scores, regression results suggest that when clinical disease is present, their effect on oculometric performance is diminished. However, consistent with other studies, years of education did persist as a significant predictor of the PHES composite score and neurocognitive tests in general [32,40]. Second, we did not perform testing on any inpatients but this is certainly worthy of study in the future, particularly in assessing dynamic changes in oculometrics as patients admitted with HE improve with treatment. Third, a few patients were unable to complete oculometric testing for various reasons including an inability to understand instructions and dense eyelashes, and as such had to repeat the test; therefore, there may have been a learning effect. Additionally, there were challenges in severely sarcopenic and debilitated patients being able to climb onto the test chair and maintain position for the duration of oculometric testing and so patients with more severe cirrhosis may be underrepresented. It is certainly plausible that the signal for our decompensated group might have been even stronger if these patients had been included. Finally, only English-speaking participants were included due to PHES language restrictions. Together, these limitations highlight the need for larger samples in which patients with cirrhosis are demographically matched to healthy controls, and a more accessible, head-mounted form factor for oculometric assessment to enable inclusion of individuals with severe cirrhosis and/or physical constraints.

Furthermore, future studies will capture information on patients who are unable to begin or complete oculometric testing to allow development of solutions that can be used to improve data collection especially for multi-site clinical studies. Future studies with larger sample sizes will be needed for more normative data collection to investigate oculometric patterns that may be associated with specific etiologies of cirrhosis. In populations with liver disease, longitudinal studies are required to assess the validity of oculometrics during disease progression to predict overt encephalopathy, falls, hospital admissions, and mortality; assessing for encephalopathy after placement of a transjugular intrahepatic portosystemic shunt (TIPS); assessing dynamic visual processing recovery after liver transplantation; and determining ability to drive in patients with cirrhosis. Similarly, longitudinal studies in patients at risk for liver disease are required to better understand the contribution of risk factors such as viral hepatitis, obesity, substance use disorder, and metabolic syndrome to neurocognitive symptoms.

The significant trend in both PHES and nFit with disease progression is consistent with the idea that routine oculometric monitoring as part of a chronic care management plan would detect functional impairments earlier than any overt clinical sign of neurocognitive decline. For diagnosis of HE, oculometric testing has several advantages as compared to the current psychometric standard of paper-and-pencil tests. Oculometric tests are easy to administer and easy for patients to follow, are largely language-free, and they provide a quick, non-invasive assessment of brain health, with sensitivity comparable to standard psychometric tests. Thus, oculometrics may fill the need for a language and education agnostic neurocognitive assessment. Scoring is automated, simple and easy to interpret, and results can be managed and accessed electronically. Detection of the early stages of HE (i.e., covert HE) will allow for earlier treatment, thereby preventing progression to later stages of HE (i.e., overt HE) and hospitalization [41], reducing healthcare expenditures and possibly even adverse outcomes like falls and motor vehicle accidents that likely stem from HE-related sensorimotor impairment, through some combination of behavioral intervention, risk management, and drug treatment. In summary, oculometric assessment captures sensorimotor impairment associated with liver disease, and after future validation studies show sufficient sensitivity compared to gold standard psychometric testing, could be used by medical assistants in the clinic to screen for HE.

## Figures and Tables

**Figure 1 geriatrics-10-00112-f001:**
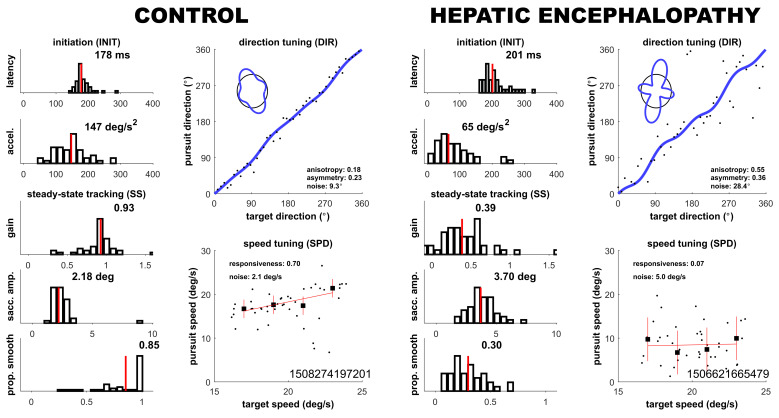
Summary of oculometric measurements for a typical control participant and a representative participant with HE in the cirrhosis group. Whereas the control participant shows responsive metrics with tight distributions of tracking metrics across trials (**left column**) and relatively little noise in direction and speed tuning (**right column**), the participant with HE shows sluggish and disorderly tracking. Inset text within the panels gives the values of each of the ten metrics.

**Figure 2 geriatrics-10-00112-f002:**
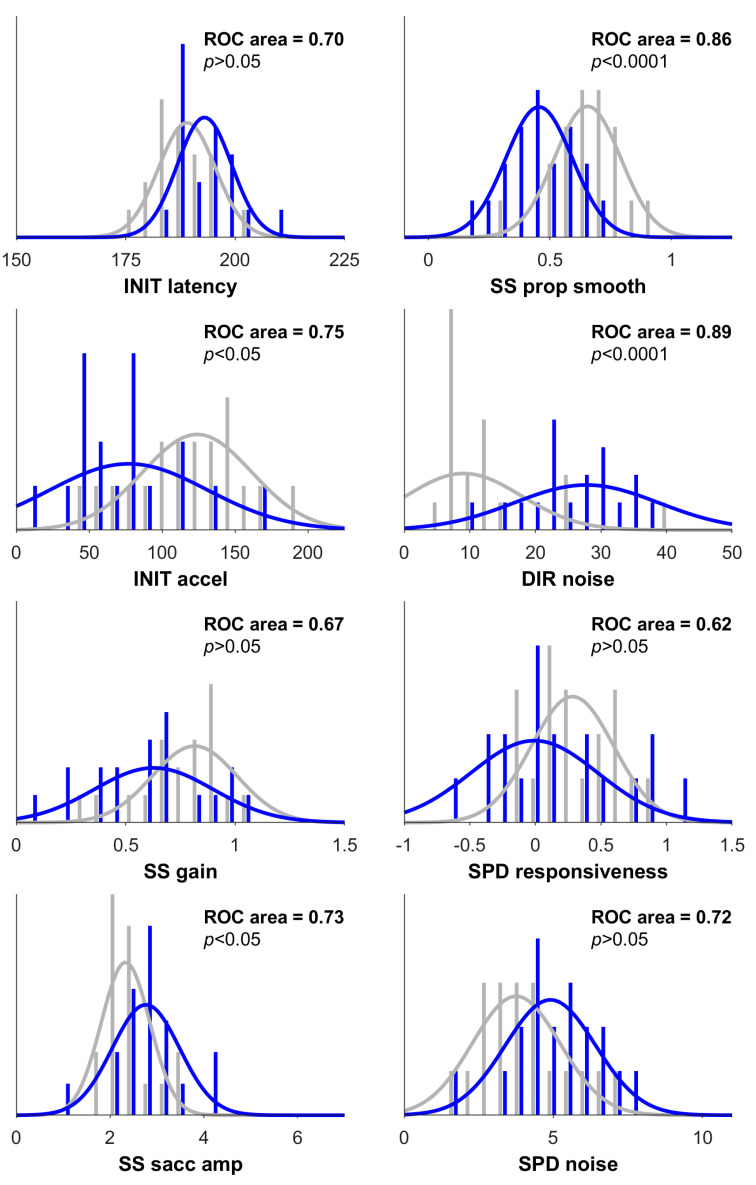
Characterization of oculometric signs associated with decompensated liver disease. Each panel shows distributions of metrics from our control (gray) and decompensated cirrhotic (blue) groups, with best-fit Gaussian distributions shown as solid lines. Sensitivity for detection of HE-related signs based upon individual metrics is inset, expressed as an ROC area. Reported *p*-values indicate the probability that the ROC area is significantly different than 0.5 by bootstrapping (10,000 samples), Bonferroni-corrected for multiple comparisons.

**Figure 3 geriatrics-10-00112-f003:**
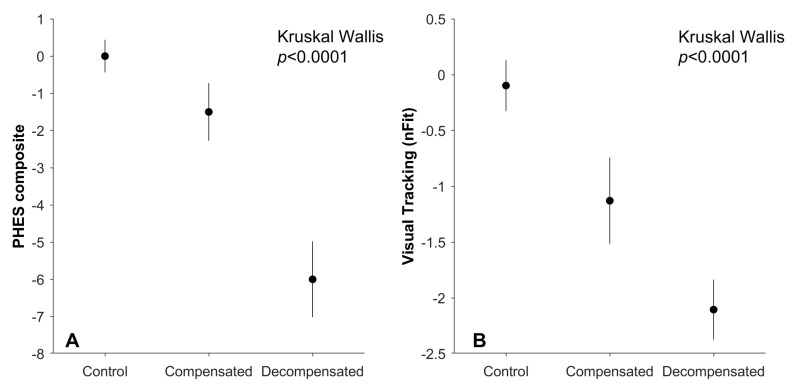
Psychometric and oculometric assessment during disease progression. Filled circles represent the median participant within each group; error bars represent standard error of the mean (SEM). Each panel plots the magnitude of functional neurological impairment (Kruskal–Wallis, *p* < 0.0001) as liver disease progresses from control to compensated cirrhosis and to decompensated cirrhosis, as measured by PHES (**A**) and nFit (**B**).

**Figure 4 geriatrics-10-00112-f004:**
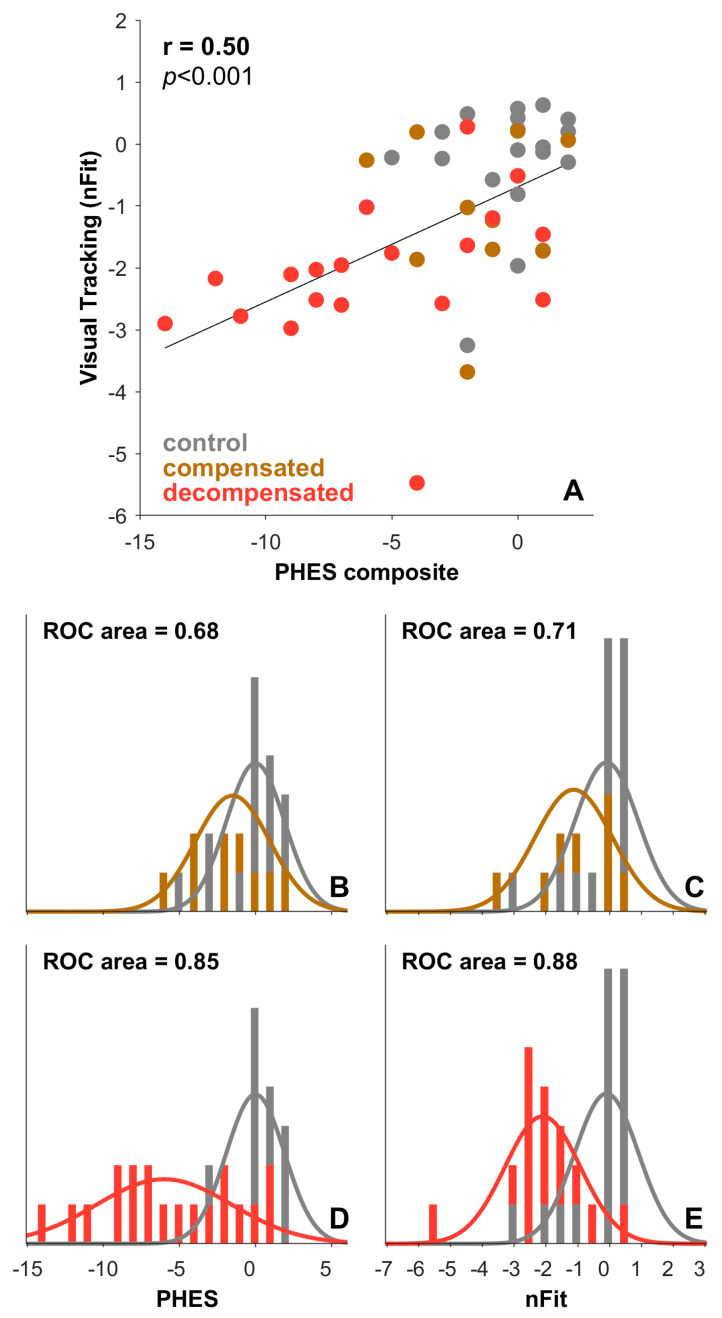
Psychometric and oculometric measurements of hepatic encephalopathy. Filled circles in (**A**) plot nFit against PHES composite for control participants (gray), and participants with compensated (yellow) and decompensated (red) liver disease. Each set of axes below (**B**–**E**) shows the sensitivity to distinguish control participants from those with cirrhosis. The solid gray histogram shows the distribution of scores from the control group; the yellow histogram shows the distribution of scores from the compensated cirrhosis group; the red histogram shows the distribution of scores from the decompensated cirrhosis group. Solid lines show the best-fit normal distribution for each histogram; the inscribed text shows the sensitivity of each test to distinguish the disease group from control. During the progression from compensated to decompensated cirrhosis, detection sensitivity increases from approximately 0.70 to 0.85 for both tests.

**Table 1 geriatrics-10-00112-t001:** Comparison of demographic data between the decompensated cirrhosis, compensated cirrhosis, and control groups, median score (standard deviation).

	Decompensated Cirrhosis (*n* = 19)	Compensated Cirrhosis (*n* = 10)	Control (*n* = 19)
Gender (% male) *	74	50	47
Age (years) ***	58 (10.16)	63 (7.42)	51 (16.65)
BMI (median) ***	29.3 (5.24)	29.2 (5.91)	23.4 (5.58)
Educational years ***	14 (3.78)	16 (2.77)	16 (1.44)
Etiology of cirrhosis (%)			
Hepatitis C Virus	26.3	60.0	N/A
NASH	26.3	20.0	N/A
Alcohol	21.1	0	N/A
Mixed	10.5	0	N/A
Other	15.8	20.0	N/A
History of hepatic encephalopathy (%)	73.6	0	N/A
History of hospitalization for HE (%)	36.8	0	N/A
Median MELD-Na score	14 (3.88)	8 (1.9)	N/A
Median CTP score	7 (1.34)	5 (0.67)	N/A

N/A: not applicable. These measures only pertain to participants with cirrhosis. * *p* < 0.05, *** *p* < 0.0001.

**Table 2 geriatrics-10-00112-t002:** Comparison of median (SD) scores on psychometric tests and oculometric measures between decompensated cirrhosis, compensated cirrhosis, and control groups.

	Decompensated Cirrhosis (*n* = 19)	Compensated Cirrhosis (*n* = 10)	Control (*n* = 19)	Kruskal–Wallis Statistic (df = 2)
PHES Composite	−6 (4.32)	−1.5 (2.33)	0 (1.87)	15.55, *p* = 0.0004
DST (# in 120 secs)	40.0 (11.2)	57 (13.6)	63.0 (12.7)	22.45, *p* < 0.0001
NCT-A (secs)	56.9 (46.3)	39.0 (13.8)	33.1 (12.4)	8.79, *p* < 0.01
NCT-B (secs)	115.6 (203.6)	75.1 (17.2)	62.2 (24.9)	20.96 *p* < 0.0001
SDT (secs)	86.5 (35.1)	78.7 (16.9)	57.7 (15.4)	11.89, *p* = 0.003
LTT (composite)	77.6 (38.7)	76.1 (41.0)	69.6 (27.1)	0.58, *p* > 0.05
MHE Dx (% yes)	63.2	30.0	5.0	16.05, *p* = 0.0003
nFit Score	−2.1 (1.2)	−1.1 (1.2)	−0.1 (1.00)	13.64, *p* = 0.001
INIT latency (ms)	193 (6)	191 (8)	189 (6)	4.21, *p* > 0.05
INIT acceleration (deg/s^2^)	77 (54)	84 (35)	124 (37)	7.25, *p* = 0.03
SS gain	0.63 (0.27)	0.58 (0.28)	0.81 (0.19)	3.17, *p* > 0.05
SS saccade amplitude (deg)	2.8 (0.7)	2.4 (0.3)	2.3 (0.5)	7.68, *p* = 0.02
SS prop smooth	0.46 (0.14)	0.56 (0.17)	0.66 (0.14)	14.57, *p* = 0.0007
DIR noise (deg)	27.8 (11.0)	18.0 (11.1)	9.1 (8.8)	14.01, *p* = 0.0009
SPD responsiveness	−0.01 (0.48)	0.46 (0.43)	0.28 (0.31)	2.87, *p* > 0.05
SPD noise (deg/s)	4.9 (1.4)	4.0 (1.4)	3.8 (1.4)	6.56, *p* = 0.04
anisotropy	2.43 (5.19)	−1.15 (2.41)	0.02 (1.92)	7.48, *p* = 0.02
asymmetry	−0.23 (4.80)	−0.72 (4.00)	−1.02 (2.58)	0.12, *p* > 0.05

DST = Digit symbol test. NCT-A = Number connection test—A. NCT-B = Number connection test—B. SDT = Serial dotting test. LTT = Line tracing test. MHE = Minimal hepatic encephalopathy. INIT = initiation. SS = steady state. DIR = direction. SPD = speed tuning.

## Data Availability

The original contributions presented in this study are included in the article/Appendix A. Further inquiries can be directed to the corresponding author(s).

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
