# Peer review of "Oculometric Assessment of Sensorimotor Impairment Associated with Liver Disease Is as Sensitive as Standard of Care Cognitive Tests"

_geriatrics, 2025, doi:10.3390/geriatrics10040112_

Round 1

Reviewer 1 Report

Comments and Suggestions for Authors

This study presents an innovative and practical approach to detecting neurocognitive impairment in patients with liver disease using oculometric testing. The authors compare a five-minute, self-administered eye-tracking assessment (nFit) to the standard psychometric hepatic encephalopathy score (PHES) and find comparable sensitivity between the two methods. The promise of a fast, non-invasive, and user-friendly test is clear and much needed in clinical hepatology, where current tools are underutilized due to their complexity.
The study design is well-structured for a feasibility project. The correlations between oculometric metrics and PHES scores are solid, and the ROC data support the claim that oculometrics can detect both compensated and decompensated cirrhosis-related impairment. The nFit’s sensitivity, particularly in identifying decompensated cases, is impressive. The figures and tables are also well-constructed and intuitive.
That said, some limitations deserve more emphasis. The groups are not well matched demographically tocontrols were younger, leaner, and more educated. which may have skewed the results. Also, patients with more severe cirrhosis might have been underrepresented due to the physical demands of the testing procedure. These factors limit generalizability and should be clearly addressed in future studies and should be discussed in the manuscript. 

Author Response

Reviewer 1Comments and Suggestions for Authors

This study presents an innovative and practical approach to detecting neurocognitive impairment in patients with liver disease using oculometric testing. The authors compare a five-minute, self-administered eye-tracking assessment (nFit) to the standard psychometric hepatic encephalopathy score (PHES) and find comparable sensitivity between the two methods. The promise of a fast, non-invasive, and user-friendly test is clear and much needed in clinical hepatology, where current tools are underutilized due to their complexity.
The study design is well-structured for a feasibility project. The correlations between oculometric metrics and PHES scores are solid, and the ROC data support the claim that oculometrics can detect both compensated and decompensated cirrhosis-related impairment. The nFit’s sensitivity, particularly in identifying decompensated cases, is impressive. The figures and tables are also well-constructed and intuitive.

 We appreciate these comments.

That said, some limitations deserve more emphasis. The groups are not well matched demographically to controls were younger, leaner, and more educated. which may have skewed the results. Also, patients with more severe cirrhosis might have been underrepresented due to the physical demands of the testing procedure. These factors limit generalizability and should be clearly addressed in future studies and should be discussed in the manuscript.

We agree and in addition to describing this demographic imbalance in lines 279 – 293 in the Discussion  section, we have emphasized these limitations in lines 311 – 331.  We also have included results from regression models in which these demographic variables are added as covariates (lines 249 – 254).

Reviewer 2 Report

Comments and Suggestions for Authors

This manuscript presents a well-executed feasibility study evaluating the use of oculometric testing as a rapid, non-invasive screening tool for hepatic encephalopathy (HE) in patients with liver cirrhosis. The study is timely and addresses a known clinical gap: current psychometric HE assessments are underutilized due to their time and resource demands. The manuscript is generally well-structured, with clear articulation of objectives, appropriate methodology, and robust statistical analysis.

Concerns:

Demographic confounders: The cirrhosis and control groups differed significantly in age, education, and BMI—all known to influence neurocognitive performance. The current analysis mentions these correlations but does not adequately control for them in group comparisons (e.g., through ANCOVA or regression modeling).

Participant selection bias: Severely impaired patients (e.g., those too debilitated for testing) may be underrepresented, possibly leading to underestimation of impairment severity. This limitation should be emphasized more clearly.

Limited generalizability: Only English-speaking participants were included due to PHES language limitations, but this restricts applicability to broader populations. The potential of oculometrics to overcome this should be highlighted more strongly as a strength.

Learning effects and testing feasibility: Some participants required repeat tests due to misunderstanding or physical difficulty. Quantification of these occurrences, and their potential influence on test-retest variability, would be helpful.

Future directions: While discussed briefly, more emphasis should be placed on the need for longitudinal validation and the exploration of predictive value (e.g., for overt HE episodes, hospitalizations, or real-world outcomes like driving safety).

Minor concerns:

Several references lack in-text citation clarity or are inconsistently cited numerically (e.g., [22], [11] not always easy to trace in the discussion).

Consider editing for minor language issues (e.g., redundancy in phrasing, such as "quick, non-invasive functional assessment" being repeated).

Figures are informative, but clearer captions and visual labels (e.g., ROC AUC numbers directly on plots) would enhance readability.

The place of reference numbers are not consistent (i.e. before or after periods/commas).

The font size of the text is not consistent.

Author Response

Reviewer 2 Comments and Suggestions for Authors

This manuscript presents a well-executed feasibility study evaluating the use of oculometric testing as a rapid, non-invasive screening tool for hepatic encephalopathy (HE) in patients with liver cirrhosis. The study is timely and addresses a known clinical gap: current psychometric HE assessments are underutilized due to their time and resource demands. The manuscript is generally well-structured, with clear articulation of objectives, appropriate methodology, and robust statistical analysis.

 We appreciate these comments.

Concerns:

Demographic confounders: The cirrhosis and control groups differed significantly in age, education, and BMI—all known to influence neurocognitive performance. The current analysis mentions these correlations but does not adequately control for them in group comparisons (e.g., through ANCOVA or regression modeling).

We now include results from regression models in which these demographic variables are added as covariates (lines 249 – 254) and have added interpretation of the regression results in the Discussion section (lines 314 – 318).  The regression models revealed that only cirrhosis diagnosis status was a significant predictor for oculometric performance, but years of education persisted as a significant predictor for PHES composite score.

Participant selection bias: Severely impaired patients (e.g., those too debilitated for testing) may be underrepresented, possibly leading to underestimation of impairment severity. This limitation should be emphasized more clearly.

 We agree and in addition to describing this demographic imbalance in lines 280 – 293 in the Discussion, we have emphasized these limitations in lines 311 – 331. 

Limited generalizability: Only English-speaking participants were included due to PHES language limitations, but this restricts applicability to broader populations. The potential of oculometrics to overcome this should be highlighted more strongly as a strength.

In the Discussion section, we now point out the language limitation of PHES on lines 327 – 328 and mention that oculometrics may fill the need for a language and education agnostic neurocognitive assessment on lines 352– 353. 

Learning effects and testing feasibility: Some participants required repeat tests due to misunderstanding or physical difficulty. Quantification of these occurrences, and their potential influence on test-retest variability, would be helpful.

 We address this issue in lines 320 – 335.  Unfortunately, the number of these occurrences is not available.

Future directions: While discussed briefly, more emphasis should be placed on the need for longitudinal validation and the exploration of predictive value (e.g., for overt HE episodes, hospitalizations, or real-world outcomes like driving safety).

 We have placed greater emphasis of this point on lines 337 – 344.

Minor concerns:

Several references lack in-text citation clarity or are inconsistently cited numerically (e.g., [22], [11] not always easy to trace in the discussion).

We have doublechecked that multiple citations are organized in numerical order.  We utilized the numbered style in Endnote which is aligned with the International Committee of Medical Journal Editors.  We note that Geriatrics has no required citation style.

Consider editing for minor language issues (e.g., redundancy in phrasing, such as "quick, non-invasive functional assessment" being repeated).

 We have doublechecked the manuscript for redundant language.

Figures are informative, but clearer captions and visual labels (e.g., ROC AUC numbers directly on plots) would enhance readability.

 We have revised the figures so that labels are more clearly visible and made the captions more clear.

The place of reference numbers are not consistent (i.e. before or after periods/commas).

 We have doublechecked the placement of reference numbers and they now consistently appear after commas and periods.

The font size of the text is not consistent.

 We have doublechecked the text for size consistency.

Round 2

Reviewer 1 Report

Comments and Suggestions for Authors

thank you for addressing my comments